# Transcriptional Expression of Nitrogen Metabolism Genes and Primary Metabolic Variations in Rice Affected by Different Water Status

**DOI:** 10.3390/plants12081649

**Published:** 2023-04-14

**Authors:** Gahyun Kim, Jwakyung Sung

**Affiliations:** Department of Crop Science, Chungbuk National University, Cheongju 28644, Republic of Korea; 1996rlark@gmail.com

**Keywords:** rice, water management, nitrogen metabolism genes, primary metabolites

## Abstract

The era of climate change strongly requires higher efficiency of energies, such as light, water, nutrients, etc., during crop production. Rice is the world’s greatest water-consuming plant, and, thus, water-saving practices such as alternative wetting and drying (AWD) are widely recommended worldwide. However the AWD still has concerns such as lower tillering, shallow rooting, and an unexpected water deficit. The AWD is a possibility to not only save water consumption but also utilize various nitrogen forms from the soil. The current study tried to investigate the transcriptional expression of genes in relation to the acquisition-transportation-assimilation process of nitrogen using qRT-PCR at the tillering and heading stages and to profile tissue-specific primary metabolites. We employed two water supply systems, continuous flooding (CF) and alternative wetting and drying (AWD), during rice growth (seeding to heading). The AWD system is effective at acquiring soil nitrate; however, nitrogen assimilation was predominant in the root during the shift from the vegetative to the reproductive stage. In addition, as a result of the greater amino acids in the shoot, the AWD was likely to rearrange amino acid pools to produce proteins in accordance with phase transition. Accordingly, it is suggested that the AWD 1) actively acquired nitrate from soil and 2) resulted in an abundance of amino acid pools, which are considered a rearrangement under limited N availability. Based on the current study, further steps are necessary to evaluate form-dependent N metabolism and root development under the AWD condition and a possible practice in the rice production system.

## 1. Introduction

The climate change is strongly raising the stakes for a paradigm shift in all industrial structures, including agriculture, and, especially, the elevating temperature and greenhouse gas emissions (GHG) are positioned as negative factors threatening sustainable food security [1]. It is forecast that a yield loss of major crops, including rice, could be an inevitable consequence, and to ensure sustainable crop production and an agricultural ecosystem, experimental approaches are constantly being attempted via expanding agricultural knowledge and practices.

Rice is one of the most important food crops, providing 20% of daily calories to more than 3.5 billion people worldwide [1]. Rice is almost exclusively produced in Asia, accounting for 90.7% of global production, and is from East Asia, where it occupies approximately 33% [2]. Unlike other crops, rice can critically grow under the flooded field environment due to the development of aerenchyma cells, which transport oxygen from the leaf to the root [3]. However, limited water irrigation owing to climate change is forcing efforts to enhance water use efficiency (WUE), and water-saving practices also lead to significant reductions in greenhouse gas emissions (CO_2_) from the rhizosphere.

One practice that is considered to achieve effective water use in rice systems is an irrigation management technique referred to as alternate wetting (saturated) and drying (unsaturated) (AWD) [4], and it was reported that an application of this practice could save water input by 23% [5] compared to continuous flooding (CF). In addition to water savings, the beneficial effect of AWD is to reduce the generation of greenhouse gases, especially methane (a 45~90% reduction) [6]. Since ammonium is a preferential form of nitrogen, adjusting water management in the rice production system may also affect nitrogen availability.

Nitrogen (N) is an essential element requiring substantially for plant growth and development, and all plant species preferentially absorb nitrogen in both inorganic and organic forms, such as nitrate, ammonium, or amino acids, from the rhizosphere. Due to the importance of nitrogen, nitrogen fertilizer has been widely used to attain better crop yields. Despite the fact that ammonium is the prevailing form in rice paddy fields, oxygen moved from the shoot via aerenchyma cells generates an aerobic environment in the rhizosphere, and, in turn, ammonium is nitrified to nitrate by nitrifying bacteria, which accounts for 40% of the gross N taken up by rice [7]. Therefore, there is evidence that rice root takes up both ammonium and nitrate, with findings as transporter genes: 84 NRTs and 12 AMTs [8]. Considering both N availability and water management in the rice production system, it could be hypothesized how the rice plant acquires and assimilates N under different water managements and, as a result, alters the primary metabolism. In addition, it remains unclear how the AWD affects N metabolism at different rice growth stages (i.e., vegetative and reproductive stages). The limited water supply perturbs the cell turgor, and, therefore, plants preferentially promote the synthesis and transport of compatible solutes, which play an essential role as osmotic protectants, such as soluble sugars and amino acids (i.e., proline and glycine-betaine) [9,10]. Thus, primary metabolites, including sugars and amino acids, are closely involved in coping with various environmental impacts.

To do this, we focused on understanding the transcriptional differences in key genes directly involved in nitrogen metabolism like uptake, transportation, assimilation, and remobilization in the CF and AWD at tillering and heading stages, and profiling targeted primary metabolites from both water management systems at the heading stage.

## 2. Results

The growth of rice plants was significantly different between continuous flooding (CF) and alternative wetting and drying (AWD), and the difference was obvious in root development (Figure 1). The AWD resulted in limited root growth (less volume), which triggered fewer tillers (17.7/plant, n = 3), compared to the CF (19.7/plant, n = 3), and slightly promoted leaf senescence. By contrast, there was no difference in the heading time. Therefore, we investigated nitrogen metabolism and metabolic alteration to see how different water management practices affected physiological responses, which caused distinct phenotypic differences.

### 2.1. Nitrogen Metabolism

The selected thirteen genes directly involved in the uptake and assimilation of nitrogen were compared between CF and AWD from the leaf blade, leaf sheath, and root of rice plants at both tillering and heading stages (Figure 2). The AWD greatly promoted the uptake of nitrate (NO_3_-N) via activating the nitrate transporter (*OsNRT2.1*) at the tillering stage, although other groups (*OsNPF2.4*, *OsNPF8.20,* and *OsAMT1;1*) were slightly expressed. Whilst the expression of OsNPF genes was obvious at the heading stage. In the AWD, an absorbed nitrate was not only quickly assimilated into ammonium in the root but also transported toward the shoot. A glutamic acid in the root was finally assimilated into glutamine by the gene expression *OsGS1;2*, encoding glutamine synthetase (GS), while the reverse reaction by *OsNADH-GOGAT1*, encoding glutamate synthase (GOGAT), was limited in the AWD compared to the CF. It was observed that nitrate-N from the root to the shoot was better transported by NRT (*OsNRT2.1* and *OsNRT2.3a*) and NPF (*OsNPF6.5*) proteins in the AWD at tillering stage. In the leaf sheath, the assimilation of nitrogen into amino acids was not significantly different between both treatments at the tillering stage; however, the expression of genes (*OsGS1;1* and *OsFd-GOGAT*) encoding GS-GOGAT metabolism was extremely reduced by the AWD treatment at the heading stage. Moreover, the expression of genes *OsNPF2.4* and *OsNPF8.2*, encoding nitrate transporter enzymes regulating nitrate influx toward leaf blades, was greatly decreased in AWD treatment at the heading stage. The gene expression encoding nitrate reductase (NR) in leaf blades was significantly enhanced by AWD treatment, whereas nitrite reductase (NIR) remained unchanged. In addition, based on the expression levels of *OsGS1;1* and *OsFd-GOGAT*, the reversibility of the GS-GOGAT pathway was higher in the AWD treatment at the tillering stage. At the heading stage, however, their expression was similar at both water managements.

### 2.2. Principle Component Analysis (PCA) of Polar Metabolites

To characterize primary metabolism in different tissues (i.e., leaf blades, leaf sheaths, and roots) by different types of water management, the targeted primary metabolites were measured using GC−TOFMS at the heading stage. The data was employed in PCA and identified major differences between water management and plant parts (Figure 3). The first two principal components explained 60.2 and 22.3% of the variability (Figure 3–top). The PC1 and PC2 clearly revealed a strong difference between leaf blade and root, and differential water management showed no effect on the PC1, indicating that the composition of targeted primary metabolites was definitely affected by plant part rather than water management. We also analyzed positive or negative effects between metabolites using loading plots (Figure 3–bottom). A majority showed a positive correlation on an *X*–axis, whereas a *Y*–axis generally separated soluble carbohydrates and organic acids from amino acids. The results implied that carbon and nitrogen metabolisms in the tissues of rice plants were greatly affected by water supply conditions.

### 2.3. Primary Metabolites

Metabolite profiles provide a much broader view of systematic adjustment in metabolic processes compared to the conventional biochemical approaches and also abundant opportunities to reveal new insights on metabolism. The targeted polar metabolites (50 biomolecules) were compared to investigate the difference between two water treatments, CF and AWD (Figure 3). The level of soluble sugars was greatly affected by tissues and water treatments. The AWD resulted in a lower level of soluble sugar pools except for xylose in leaf blades, whereas levels were significantly higher in the leaf sheath and root. In contrast, glucose-6-P and fructose-6-P in the root were significantly reduced by the AWD. A mannitol, of sugar alcohol, was greatly accumulated in the AWD−grown root. Most amino acids showed a tendency for accumulation by the AWD, and the significantly higher level of glutamine and asparagine was notable, indicating 1.0- and 1.4-fold (log_2_ scale) greater in leaf blades compared to CF and 4.2- and 6.8-fold higher in the leaf sheath, respectively. Especially, the obviously accumulated amino acids in the leaf sheath were valine (2.3-fold), serine (2.3), leucine (2.3), isoleucine (3.0), proline (2.5), glycine (2.5), threonine (2.3), β-alanine (3.4), methionine (3.3), lysine (2.6), tryptophan (3.9), and putrescine (2.2), including glutamine and asparagine in the AWD treatment. In AWD-treated roots, the level of putrescine was markedly increased. A majority of organic acids, including the TCA intermediates, were not significantly different from the water treatments, whereas pyruvic and fumaric acids were greatly accumulated in leaf blades.

## 3. Discussion

The current work has been studied to better understand a variation in nitrogen availability and primary metabolism under water−saving management (alternate wetting and drying, AWD), and is discussed with an emphasis on the significant findings. It has been widely verified that limited water supply contributed to the lower biomass production as a result of the trade−off between growth and stress defense [11]. The observations of root development under limited water supply differed from experimental conditions such as pot and/or field. Root development tended to increase as a result of the water stress response [12,13]. On the other hand, it was slightly affected by limited water [14] or shallower [15]. Our results showed that the AWD led to a reduction in biomass, tillering, and root development, which was partly in line with previous observations.

The transcriptional expression of selected genes functioning in the acquisition, transportation, assimilation, and remobilization of nitrogen in the leaf blade, leaf sheath, and root of the rice plant was compared between CF and AWD at both tillering and heading stages. We found that the uptake of nitrate was obviously promoted by the AWD, and, especially, it was dominant in high (*OsNRT2.1*) and low (*OsNPF2.4*, nitrate peptide family) affinity transporters at the tillering stage. In contrast, the dual affinity transporter of nitrate, *OsNPF6.5*, was significantly upregulated at the heading stage. The upregulation of *OsNRT2.1* enhanced nitrate uptake to increase assimilation efficiency in an AWD−employed rice system [16]. By contrast, a drought significantly decreased the expression of *OsNRT2.1* [16]. From our study, both NRT and NPF genes were positively promoted under moderate water limitation (AWD). Thus, it is implied that AWD practice might contribute to enhanced availability of nitrate and, finally, improved N use efficiency (NUE). Another very interesting finding was that the difference in N assimilation strongly depended on the growth stage, tillering vs. heading, in the AWD. At the tillering stage, soil N was preferentially transported to the shoot to be assimilated into amino acids, whereas at the heading stage, root N utilization was dominant.

Photosynthetic rate was not significantly affected under mild and/or moderate soil drying due to enhanced N assimilation in the rice leaves [17], indicating a collaboration of photosynthesis and N metabolism under limited watering. Matsunami et al. (2018) [18] reported that the expression of N metabolism genes (*OsAMT1;3*, *OsAMT2;1*; *OsGS1;2*, *OsGOGAT2*) was greater at heading than at tillering, probably reflecting long−term N demand during the growth and development of the rice plant. Reduced osmotic potential at reproductive stage resulted in a decrease in the expression of NR, NIR, and GS-GOGAT genes in flag leaves of the rice plant [19] due to inhibition of water and nitrate movement at root level [20,21]. Our results implied that the uptake and assimilation of nitrogen under moderately limited water conditions (e.g., AWD) greatly differed depending on the growth stage. At the vegetative stage, acquired nitrogen is primarily utilized to increase photosynthesis and leaf biomass, whereas at the reproductive stage, it is likely to be consumed to promote root development as a countermeasure to extend the soil N−acquiring zone. Further investigation is also required to clarify that an expansion of the rhizosphere via upregulated expression of nitrogen metabolism genes is closely involved in the development of panicles under moderate water limitation (AWD). 

Non-structural soluble carbohydrates (NSCs) are especially important within a group of osmoprotectants. Soluble carbohydrates, including sucrose, glucose, and fructose, are greatly accumulated in tissues during limited water conditions like drought [22,23,24,25,26]. An abundance of soluble sugars measured at the heading stage greatly differed from tissues, with a tendency to decrease in source (leaf blade) and increase in sink (leaf sheath and root). In particular, the level of glucose and fructose was significantly accumulated in the root in AWD. Our results suggested that the transportation from source to sink was considered (1) a typical response to water stress, (2) a temporary retention in the leaf sheath for transporting to grain (reproductive tissue), and/or (3) a promotion of root development to extend the water-/nutrient-acquiring zone. The perturbation of soluble sugars by limited water supply somewhat differed from plant species and a period of water limitation. Water stress, like drought, resulted in an accumulation of soluble sugars, including sucrose, in plant roots [27,28,29]. However, a short−term water limitation led to a decrease in glucose and fructose in a leaf of a cereal crop, whereas an increase in xylose [25]. In contrast, an accumulation of soluble sugars, including sucrose, remained unchanged under limited water conditions [23,30]. Therefore, we carefully suggest further work to understand the variation of carbohydrate metabolism during the vegetative to reproductive transition under the AWD.

A higher level of amino acids reflects sensitivity due to the increased degradation and decreased synthesis of proteins in a water−limited environment [31,32]. The significant accumulation of amino acids from our study, especially in leaf blade and sheath, may be closely associated with nitrogen storage for further metabolic remobilization [33] and was clearly explained with down−regulation of transportation/assimilation−involved genes (Figure 1). Indeed, the AWD resulted in higher ratios of glutamine (gln)/glutamate (glu) and asparagine (asn)/aspartate (asp) in both the leaf blade and sheath. This agrees with the findings that water limitation accumulated not only higher levels of glutamine and asparagine but also other amino acids, including glycine, valine, and alanine [23,25,34,35,36]. Accordingly, the ratio of gln/glu and asn/asp could be a reliable biochemical marker to evaluate N metabolism like the NUE in the AWD. Previous studies have highlighted that plants experiencing water stress accumulate proline [37,38,39], an osmotic adjuster, and γ−aminobutyric acid (GABA) [40], a stress marker. We also observed that proline and GABA showed significantly higher increases in the AWD compared to CF. A higher level of those amino acids in the roots indicates that the AWD−grown rice plants might experience moderate water stress. Polyamines are closely linked with proline in terms of their biosynthetic and catabolic pathways [41] and play a role in mitigating abiotic stresses such as drought [42]. The AWD showed a significant increase in the level of putrescine in roots. Many reports showed the opposite behavior of putrescine. Accumulating putrescine in rice [43] and maize [44] is a tolerant response to drought. On the other hand, its level decreased in rice [45,46] and tomato [47] during the drought, helping those sensitive organisms accumulate putrescine [48]. Therefore, it is carefully proposed that an abundance of putrescine in our study is a collaborative tolerance mechanism with proline and GABA under an intermittently water−limited environment such as the AWD.

## 4. Materials and Methods

### 4.1. Plant Material, Growth Condition, and Treatment

This study was conducted at an experimental greenhouse of Chungbuk National University, Cheongju, Republic of Korea (36°37′48.6″ N, 127°27′05.3″ E) from April to August 2021. The soil was sandy loam with 6.2 of pH, 0.28 dS/m of EC, 0.1 g kg^−1^ of total nitrogen, 106 mg kg^−1^ available P_2_O_5_, 0.61 cmol^+^ kg^−1^ of potassium (K), 2.7 cmol^+^ kg^−1^ of calcium (Ca), 1.38 cmol^+^ kg^−1^ of magnesium (Mg), and 5.6 cmol^+^ kg^−1^ of CEC. 

The seeds of rice (Oryza sativa cv. Jinbongbyeo) were sterilized for 48 h at 28 °C in 5 L of water, including 2.5 mL of seed sterilizer (Kimaen, Farm Hannong), transferred to an incubator (28 °C, darkness) for 5 days, then moved to a growth chamber (VS-91G09M-2600, Vision Scientific, Republic of Korea) with a 12/12 h photoperiod, 60% (*w*/*v*) relative humidity (RH), and 24/20 °C (day/night) after de-etiolation, in order to induce uniform germination. Uniformly growing rice seedlings (3rd to 4th-leaf stage) were transplanted into plastic containers (20 × 20 × 20 cm), including the soil.

N (urea), P (superphosphate), and K (KCl) were applied at a rate of 90-45-56 kg/ha (a standard fertilization recommendation, Rural Development Administration, Republic of Korea). N and K were split into three stages (50-30-20% at basal−tillering−panicle differentiation) and two stages (60-40% at basal−panicle differentiation), respectively. Two different types of water management were employed for this study: (1) constant flooding (CF, 5 cm height from the topsoil surface) as the control group, and (2) alternative wetting and drying (AWD, 0~5 cm of water level by evapotranspiration). Water supply for all treatments was temporarily ceased during the non−productive tillering stage (no panicle tiller). The rice plants (n = 3, each treatment) were taken at tillering (approximately 30 days after transplanting) and heading (panicle emergence) stages, immediately washed out with ddH_2_O, divided into leaf blades, leaf sheaths, and roots, and stored at −80 °C for further molecular and biological analysis.

### 4.2. RNA Extraction, cDNA Synthesis, and Relatively Quantitative Real−Time Polymerase Chain Reaction (qRT−PCR)

Total RNA was extracted from leaf blades, leaf sheaths, and roots of rice at the tillering and heading stages, using TRIzol reagent (Invitrogen, Carlsbad, CA, USA), according to the manufacturer’s instructions. The extracted RNA purity and concentration were measured by NanoDrop (Thermo Fisher Scientific, Madison, WI, USA) and double−checked by electrophoresis on a 1% agarose gel. Complementary DNA (cDNA) was synthesized by using a Maxime RT PreMix Kit with oligo (dT) primers and 1 μg of total RNA. Quantitative Real−Time PCR was performed using SYBR Green Q Master mix ROX (SYBR) (LaboPass, Seoul, Republic of Korea) with a CFX Connect Optics Real−Time System (Bio-Rad, Hercules, CA, USA) according to the manufacturer’s instructions. To analyze gene expression, put 1 μL of cDNA, 2 μL of forward primer (FW), 2 μL of reverse primer (RV), and 5 μL of SYBR in a PCR tube. Quantification method (2^−ΔΔCt^) was used, and rice actin primer (*OsACT-1*, *Os03g0718100*, FW: 5′-TGTATGCCAGTGGTCGTACC-3′, RV: 5′-CCAGCAAGGTCGAGACGAA-3′) was used for the control gene to normalize the data. The expression of selected genes (Appendix A) was validated using four biological replicates and two technical replicates. The PCR program consisted of initial denaturation at 95 °C for 10 min, followed by 40 cycles of denaturation at 95 °C for 20 s, annealing at 50–59 °C for 30 s, and extension at 72 °C for 30 s. After the last extension step, perform the melt curve step at 65 °C for 5 s and 95 °C for 0.5 s.

### 4.3. Extraction and Analysis of Polar Compounds 

Polar compounds (e.g., sugars, sugar alcohols, amino acids, organic acids, and secondary metabolism intermediaries) were analyzed in accordance with a previously reported method [16]. The powdered samples (100 mg) were placed in microtubes. 1 mL of a methanol:water:chloroform (2.5:1:1, *v*/*v*/*v*) [17] solution was added to the tubes. Used ribitol (60 μL, 0.2 mg mL^−1^) solution as an internal standard (IS). After vortexing it, the mixture was incubated at 37 °C and subjected to 1200 rpm for 30 min in a thermomixer (Eppendorf AG, Hamburg, Germany). was After centrifuging the mixture for 5 min at 4 °C and 16,000× *g*, 0.8 mL of the supernatant was transferred to new tubes, and 0.41 mL of deionized water was added. Centrifuged the mixture at the same conditions (4 °C, 16,000× *g* for 5 min). After that, 0.9 mL of the supernatant was transferred to new tubes and dried in a vacuum concentrator (VS-802F; Visionbionex, Gyeonggi, Republic of Korea) for at least 3 h, and then freeze−dried for 16 h. For derivatization, add 80 μL of methoxyamine hydrochloride (20 mg mL^−1^) to pyridine and shake at 30 °C for 90 m. To execute trimethylsilylated etherification, 80 μL of N-methyl-N-(trimethylsilyl) trifluoroacetamide (MSTFA) was added to the tubes and incubated at 37 °C for 30 min. The samples were analyzed on a GC (Agilent 7890A, Agilent Technologies, Santa Ckara, CA, USA) equipped with a Pegasus TOF−MS (LECO, St. Joseph, MI, USA). A Rtx-5MS column (30 cm × 0.25 mm, 0.25-μm i.d. film thickness; Restek, Bellefonte, PA, USA) was used to separate the polar compounds. The oven temperature was programmed as follows: 80 °C for 2 m, followed by ramping to 320 °C at 15 °C/min and holding at this temperature for 50 min. Helium gas (He) was passed at a rate of 1 mL min^−1^, and 1 μL of sample was injected at a 1:25 ratio in split mode. The inlet temperature was set at 230 °C, respectively, and spectral data were scanned over an m/z mass range of 85 to 600. Data were analyzed using ChromaTOF software (v5.5; LECO), and peaks were identified based on mass−spectral data compared to standards (i.e., NIST 11, Wiley 9, and in−house libraries). For quantification, the relative ratio of the peak area of the compounds to the peak area of the IS was acquired based on the selected ions.

### 4.4. Statistical Analysis

Statistical analysis was performed by the RStudio (version R−4.1.3, RStudio Team, 2022). The qRT−PCR data was subjected to a *t*-test. Polar compounds were analyzed by principal component analysis (PCA), subjected to a *t*-test, and visualized by a heatmap. 

## 5. Conclusions

The current study aimed to extend our knowledge of carbon (primary metabolites) and nitrogen (acquisition and assimilation) metabolism under limited water supply (alternative wetting and drying, AWD) during rice growth and development. The AWD system promoted not only the acquisition of nitrate but also its transportation and assimilation, and, in particular, nitrogen assimilation was preferential in the root during the shift from the vegetative to reproductive stage. Based on the significant increase in amino acids in the shoot (leaf blade and sheath), the AWD was considered to rearrange amino acid pools in order to synthesize proteins in accordance with phase transition. AWD acquired more active nitrate from soil and resulted in an abundance of amino acid pools, which are considered a rearrangement for reproduction−related proteins under limited N availability. We have taken some fruitful information to provide directions for further study to clarify form−dependent N metabolism and root development under the AWD condition and a practical approach in the rice production system.

## Figures and Tables

**Figure 1 plants-12-01649-f001:**
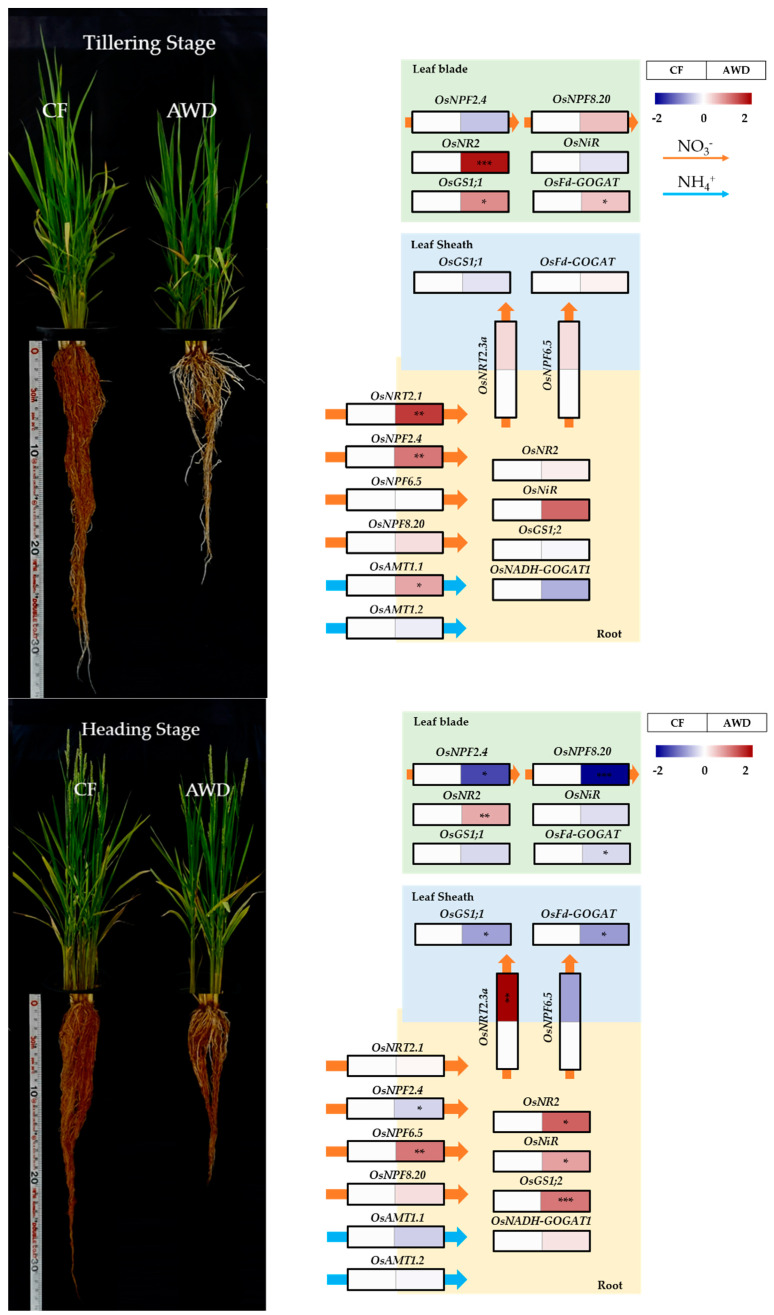
Rice (*O. sativa* cv. Jinbongbyeo) growth at both tillering and heading stages. The expression of genes (relative level of AWD to CF) involved in nitrogen metabolism in leaf blade, leaf sheath, and root at both tillering and heading stages subjected to different water managements; CF, continuous flooding; AWD, alternative wetting and drying (n = 3). The colors red and blue represent positive and negative expressions of genes, respectively. The qRT−PCR data was subjected to a *t*-test. *, **, and *** indicate the significance at *p* < 0.05, 0.01, and 0.001, respectively.

**Figure 2 plants-12-01649-f002:**
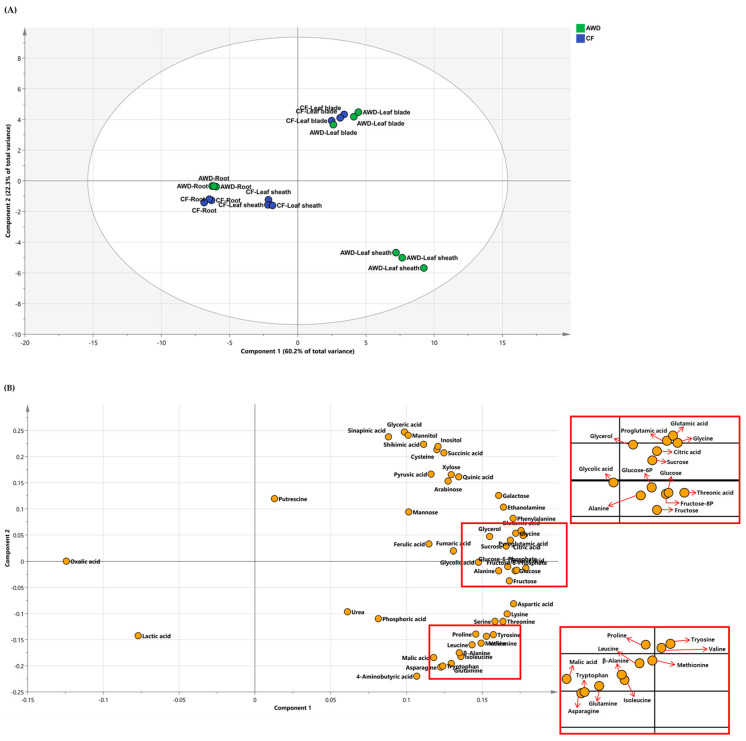
Principal component analysis (PCA) score (**A**) and loading (**B**) plots of metabolites identified from leaf blades, leaf sheaths, and roots of rice plants at heading stage.

**Figure 3 plants-12-01649-f003:**
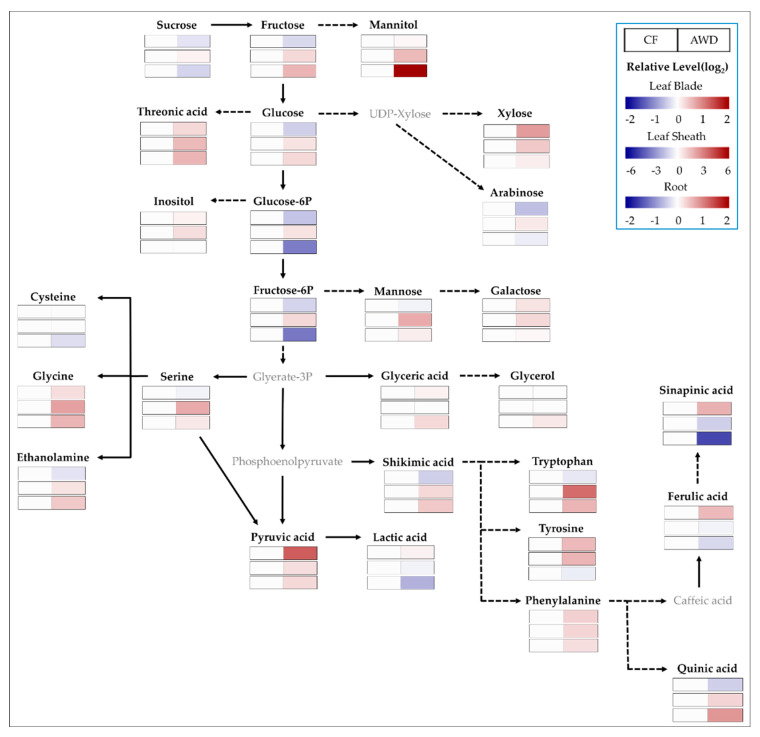
Primary metabolic changes (relative level of AWD to CF) in leaf blade, leaf sheath, and root at heading stage subjected to different water managements; CF, continuous flooding; AWD, alternative wetting and drying (n = 3).

## Data Availability

Not applicable.

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
