# Peer review of "Transcriptional Expression of Nitrogen Metabolism Genes and Primary Metabolic Variations in Rice Affected by Different Water Status"

_plants, 2023, doi:10.3390/plants12081649_

Round 1

Reviewer 1 Report

Review 27.03.2023 Plants

In the manuscript titled “Transcriptional Expression of Nitrogen Metabolism Genes and Primary Metabolic Variations in Rice Affected by Different Water Status” the authors investigated the effect of alternative wetting and drying (AWD) regime on the expression of key genes involved in nitrogen metabolism of rice at different stages of its development, as well as the composition of tissue-specific primary metabolites (50 biomolecules).  They showed that AWD rearranged amino acid pools due to the differential activity of the nitrogen related genes at heading and tillering stages.  They also found that the levels of soluble sugars and organic acids did not depend significantly on the water regime, in contrast to most of amino acids. The study adds valuable information on nitrogen availability and primary metabolism under water-saving management but I think the manuscript cannot be accepted for publication in its present form.

Major comments:

First of all, I would recommend intensive editing by a native speaker, as the manuscript is replete with grammatical errors, which sometimes make it difficult to understand the meaning of the text.

A sample of three plants for each type of treatment appears to be quite modest even for preliminary conclusions.

The authors claim that AVD "stimulates the biosynthesis of proteins responsible for reproduction" (line 315). They also state that the content of soluble sugars does not change with AWD (lines 130, 131). How do these assumptions correlate with limited biomass production and productivity (lines 161, 162).  Perhaps  determination of stress indicators could contribute to a better understanding of this discrepancy?

In general, the experimental data need to be carefully discussed. In the discussion the data obtained by the authors are interspersed with the citations from similar studies. I would recommend a more detailed analysis of the authors' own results and literature data

Minor issue:

There are no references to the table S1 in the text of the manuscript.

Author Response

Thank you for your kind comments and suggestions.

Please find the attached.

Once again, thank you for your dedication.

Reviewer 2 Report

The manuscript contains several sentences that are difficult to understand. I suggest improving grammar and syntax to clarify the reading.

I have some observations:

Line 12 and 15  What is the meaning of etc. Would you like to specify?

Line 24

 derives the reproduction-responsible 24 protein rearrangement.

It is not clear what it means. Please elaborate.

Line 25 and 26. This paragraph does not explain the results obtained. Could broadly elaborate on the meaning of the results and future work?

I am curious, why the roots of ADW plants are darker than the CF?

Author Response

(The authors gave the same response as above.)

Round 2

Reviewer 1 Report

Review 2

The changes made to the manuscript undoubtedly contributed to its improvement. However, I would again strongly advise editing by a native speaker. In particular, in my opinion:

1. lines 161-162: in should be added

2. line 169: in should be removed

3. "from” should be replaced by “depending on the growth stage”

4. line 195: insert depending on tissues

5.line 220: replace adjustment with adjuster

6. lines 249-250: “N and K 248 were split with three stages (50-30-20% at basal-tillering-panicle differentiation) and two 249 stages (60-40% at basal- tillering-panicle differentiation), respectively.”

 Please, clarify which two stages are in question in the second case

I noted the shortcomings that distort the meaning only in the corrections made , without risking giving recommendations throughout the text

Author Response

Dear Reviewer,

Thank you for your kind comments.

I totally agree with you and revised according to your request.

The revised was expressed as red color in a manuscript.

    - lines 161-162: in should be added - Added

    - line 169: in should be removed -Removed

    - "from” should be replaced by “depending on the growth stage” - Changed

    - line 195: insert depending on tissues - Inserted

    - line 220: replace adjustment with adjuster - Changed

    - lines 249-250: “N and K 248 were split with three stages (50-30-20% at basal-tillering-panicle differentiation) and two 249 stages (60-40% at basal- tillering-panicle differentiation), respectively.” - 0-40% at basal-panicle

Once again, thank you for providing your precious time and dedication.
